# Epigenetic Effects of Gender-Affirming Hormone Treatment: A Pilot Study of the ESR2 Promoter’s Methylation in AFAB People

**DOI:** 10.3390/biomedicines10020459

**Published:** 2022-02-16

**Authors:** Francesco Pallotti, Giulia Senofonte, Fani Konstantinidou, Silvia Di Chiano, Fabiana Faja, Flavio Rizzo, Francesco Cargnelutti, Csilla Krausz, Donatella Paoli, Andrea Lenzi, Liborio Stuppia, Valentina Gatta, Francesco Lombardo

**Affiliations:** 1Laboratory of Seminology—Sperm Bank “Loredana Gandini”, Department of Experimental Medicine, University of Rome “Sapienza”, 00161 Rome, Italy; francesco.pallotti@uniroma1.it (F.P.); giulia.senofonte@uniroma1.it (G.S.); silvia.dichiano@uniroma1.it (S.D.C.); fabiana.faja@uniroma1.it (F.F.); flavio.rizzo@uniroma1.it (F.R.); francesco.cargnelutti@uniroma1.it (F.C.); donatella.paoli@uniroma1.it (D.P.); andrea.lenzi@uniroma1.it (A.L.); francesco.lombardo@uniroma1.it (F.L.); 2Department of Psychological Health and Territorial Sciences, School of Medicine and Health Sciences, “G. d’Annunzio” University of Chieti-Pescara, 66100 Chieti, Italy; fanikonst@hotmail.com (F.K.); stuppia@unich.it (L.S.); 3Unit of Molecular Genetics, Center for Advanced Studies and Technology (CAST), “G. d’Annunzio” University of Chieti-Pescara, 66100 Chieti, Italy; 4Department of Experimental and Clinical Biomedical Sciences “Mario Serio”, University of Florence, 50139 Florence, Italy; csilla.krausz@unifi.it

**Keywords:** GAHT, testosterone, methylation, *ESR2*, gender incongruence

## Abstract

Virilization of gender-incongruent subjects to whom were assigned the female gender at birth (AFAB) is achieved through testosterone administration. Inter-individual differences in the timing and acquisition of phenotypic characteristics, even if the same hormone preparations and regimens are used, are frequently observed. Polymorphisms of sex hormone receptors and methylation of their gene promoters, as well of several imprinted genes as H19, may underlie the differential response to treatment. Thus, the aim of this study was to examine the possible relationship between the CpG methylation profile of the estrogen receptor 2 gene (*ESR2*) and *H19* promoters and their influence on phenotype modifications in a cohort of AFAB people at baseline (T0) and after 6 mo (T6) and 12 mo (T12) of testosterone therapy (testosterone enanthate, 250 mg i.m. every 28 d). A total of 13 AFAB subjects (mean age 29.3 ± 12.6) were recruited. The percentage of methylation of the *ESR2* promoter significantly increased at T6 (adj. *p* = 0.001) and T12 (adj. *p* = 0.05), while no difference was detected for *H19* (*p* = 0.237). Methylation levels were not associated with androgen receptor (AR)/estrogen receptor beta (ERβ) polymorphisms nor hormone levels at baseline and after six months of treatment. On the other hand, total testosterone level and patient age resulted in being significantly associated with *ESR2* methylation after twelve months of treatment. Finally, the difference in *ESR2* promoter methylation between T6 and baseline was significantly associated with the number of CA repeats of the ERβ receptor, adjusted vs. all considered variables (R^2^ = 0.62, adj. R^2^ = 0.35). No associations were found with CAG repeats of the *AR*, age, and estradiol and testosterone levels. Despite the small sample size, we can hypothesize that treatment with exogenous testosterone can modify the *ESR2* methylation pattern. Our data also indicated that epigenetic changes may be regulated, suggesting that the modulation of estrogen signaling is relevant shortly after the beginning of the treatment up to T6, with no further significant modification at T12. Furthermore, estrogen receptor methylation appears to be associated with the age of the subjects and exogenous testosterone administration, representing a marker of androgenic treatment. Nonetheless, it will be necessary to increase the number of subjects to evaluate how epigenetic regulation might play a relevant role in the modulation of phenotypical changes after testosterone treatment.

## 1. Introduction

Gender-affirming hormonal therapy (GAHT) is critical for phenotypical and psychological transition in adults with gender incongruence (GI), which is an enduring and marked incongruence between the individual’s desired gender and the assigned sex at birth. While this definition includes a broad spectrum of transgender people (stretching between binary and non-binary subjects), the current scientific literature generally refers to assigned females at birth (AFAB) and assigned males at birth (AMAB) subjects [1].

Transgender people are often considered a psychologically vulnerable group compared to the general population, partly due to distress deriving from GI itself and partly to a non-inclusive cultural background, which might also include difficulties in accessing general healthcare and GAHT [1,2]. GAHT, in particular, is a paramount tool to alleviate the incongruence between biological and experienced gender and, therefore, any psychological burden arising because of it. Moreover, physical changes induced by a hormone gender-affirming treatment are associated with an improvement of the subjects’ quality of life and self-esteem, reducing anxiety, depression, and social distress [3].

This is achieved through the reduction of endogenous sex hormones and their replacement with exogenous hormones, allowing the progressive substitution of the incongruent patient’s secondary sexual characteristics with the desired ones. In virilizing therapy for incongruent subjects who have been assigned the female gender at birth (AFAB people), the treatment of choice relies on one of the many testosterone preparations available, following the principles of hormone replacement therapy in hypogonadal patients [4]. Cessation of menses, clitoridomegaly, increased muscle/fat mass ratio, facial and body hair growth, voice deepening, and other desired features usually appear within months to years. Nonetheless, it has been observed that inter-individual differences in the timing and acquisition of phenotypic characteristics are frequent, even in subjects who undergo the same hormone preparation and regimen. There are several explanations for this phenomenon, many of which are linked to inter-individual differences in hormone signaling due to the polymorphisms of sex hormone receptors [5]. It is likely that hormone receptor activity might be modulated by both congenital and acquired factors. Unfortunately, literature data on genetic and epigenetic contributions in this setting are lacking. Nonetheless, gene promoters are capable of being activated or inhibited by physiological changes or by the action of exogenous compounds with estrogenic activity [6]. The most frequently characterized epigenetic modification is DNA methylation, which involves inherent and acquired gene transcription changes, independent of the DNA sequence.

Scant data are available for the androgen receptor (*AR*), and a few authors have investigated the occurrence of modifications of the estrogen receptor 1 gene (*ESR1*) promoter methylation after GAHT [7,8]. The functional biological meaning of the induced methylation pattern is still unknown, but it is possible to speculate that *AR* and *ESR1* genes’ methylation could play a role in modulating the induction of desired phenotypical changes, as well as being a putative marker of hormonal treatment. On the other hand, no data are available on the estrogen receptor 2 (*ESR2*) gene.

Moreover, *H19* is a paternal imprinted gene whose hypomethylation is associated with the Silver Russell syndrome and other anomalies (pre-natal and post-natal growth deficit, insulin resistance, polycystic ovarian syndrome, and precocious pubarche) [9]. Nonetheless, DNA methylation of *H19* has also been associated with pubic hair onset and genital or breast development in adolescent boys and girls, respectively [10]. Thus, the *H19* promoter methylation could be potentially related to a differential development of secondary sex characteristics after GAHT.

Thus, the main purpose of this study was to examine the possible relationship between the CpG methylation profile of *ESR2* and *H19* promoters and their possible influence on acquired changes in a cohort of AFAB people at baseline (T0) and after 6 mo (T6) and 12 mo (T12) of testosterone therapy.

## 2. Materials and Methods

We prospectively recruited consecutive AFAB people referring to the Gender Incongruence Ambulatory of the Department of Experimental Medicine (AOU Policlinico Umberto I—“Sapienza” University of Rome) before administration of GAHT. The diagnosis of gender dysphoria was confirmed by mental health specialists (Servizio di Adeguamento tra Identità Fisica e Identità Psichica—SAIFIP, AO San Camillo Forlanini), according to the DSM-V criteria. All patients provided their informed consent before the start of the treatment and the study was approved by Ethics Committee “Sapienza” (Ref. 6554; Protocol 1057/2021; 30 November 2021). Each patient underwent physical examination, Ferriman–Gallwey scoring [11], and hormonal blood testing before and after the administration of testosterone enantate 250 mg/mL i.m. once every 28 d. Additional hematological and biochemical data were retrieved from patient’s medical records.

### 2.1. Hormone Profile

Each recruited subject provided a peripheral blood sample (8 a.m.) after overnight fasting. Serum follicle-stimulating hormone (FSH), luteinizing hormone (LH), and testosterone were quantified by Chemiluminescent Microparticle ImmunoAssay (CMIA, Architect System; Abbott Laboratories, Abbott Park, IL, USA). Detection limits, intra- and inter-assay coefficients of variation, as well as normal ranges from our laboratory were previously described [12].

### 2.2. DNA Extraction and AR and ER Polymorphisms

DNA was extracted from peripheral blood leukocytes using the Wizard Genomic DNA Purification Kit (Promega, Madison, WI, USA). Extracted DNA was quantified by NanoDrop ND-2000 (Thermo Fisher Scientific, Waltham, MA, USA). For the androgen receptor, located on the X chromosome, the analysis of the percentage of the inactivation of the alleles was performed as proposed by Zitzmann et al. (2004) [13]. Briefly, methylation-sensitive restriction enzyme HpaII (Promega, Madison, WI, USA) was used to digest non-methylated DNA, hence allowing the selective PCR amplification of methylated (inactive) DNA regions. Genomic DNA (1 µg) was digested for 6 h at 37 °C with 20 U of HpaII, according to the manufacturer’s instructions, followed by incubation at 65 °C for 30 min to inactivate the restriction enzyme. Amplification for both digested and non-digested samples was performed for a volume of 25 μL (0.5 ng gDNA, 0.8 μM of each primer, 12.5 μL of Ampli Taq Gold 360 Master Mix—Applied Biosystems, Carlsbad, CA, USA).

The percentage of X chromosome inactivation was then calculated using Gene Mapper Analysis software v. 4.1 (Applied Biosystem, Carlsbad, CA, USA). Variables were defined as: digested allele 1 (A), digested allele 2 (B), non-digested allele 1 (C), and non-digested allele 2 (D). Normalization of the results from the two independent reactions and compensation for iniquitous allele amplifications due to confounding factors other than methylation were achieved through a correction factor [13]:Allele 1 inactivation: (A/C)/(A/C + B/D)(1)
Allele 2 inactivation: (B/D)/(A/C + B/D)(2)

The extreme values of 0 and 1 correspond to no inactivation and complete inactivation, respectively. A value of 0.5 was interpreted as a random inactivation. The sum of inactivation percentages of alleles 1 and 2 must be equal to 1 (Equation (1) + Equation (2) = 1). To determine the length of the polymorphic fragments in the androgen receptor (AR) and estrogen receptor beta (ERβ) genes (CAG and CA repeats, respectively), DNA samples were analyzed by linear sequencing (3500 Genetic Analyzer, Applied Biosystems) using primers flanking the triplets’ repeat regions. The forward primer for CA and CAG was fluorescently labelled with FAM at 5′ to enable the fragment to be seen during electrophoresis (fragment analysis) [14]. In Appendix A shows the primers used for the CA and CAG analyses. Raw data from the capillary electrophoresis were analyzed by Gene Mapper Analysis (Applied Biosystems).

### 2.3. Epigenetic Analysis

One-hundred nanograms of DNA was bisulfite-converted using the EpiTect Plus DNA Bisulfite Kit (Qiagen). PCR amplification of the *ESR2* and *H19* promoters was performed as previously described by different authors [15,16] Bisulfite-converted DNA served as the template for the polymerase chain reaction (PCR) followed by pyrosequencing (PSQ). The PCR mix included PyroMark PCR Master Mix, 2×, and CoralLoad Concentrate, 10× (Qiagen), 0.2 µM of each primer, 1 µL of converted DNA, and nuclease-free water to a final volume of 25 µL. Primers used for DNA methylation analysis and PCR cycling conditions are shown in Table 1. The pyrosequencing reaction was run on a PyroMark Q96ID (Qiagen), and CpGs methylation analysis was conducted by the PyroMark CpG software (Qiagen). The methylation for each amplicon was calculated as the median of the methylation status of each analyzed CpG.

### 2.4. Statistical Analysis

Continuous variables are presented as the mean ± SD or the median and IQR, depending on the shape of the distribution curve evaluated by the Kolmogorov–Smirnov test. Categorical variables are presented as counts and percentages. Differences in methylation pattern across samples were calculated by Friedman’s test. A two-tailed *p*-value of 0.05 was considered significant. Associations between methylation and the other evaluated parameters were evaluated through linear regressions and generalized linear models. All computations were carried out with the Statistical Package for the Social Sciences (SPSS) 25.0 (SPSS Inc., Chicago, IL, USA).

## 3. Results

A total of 13 AFAB subjects (mean age 29.3 ± 12.6, median 21 (IQR 19.5–43.5) y) provided a blood sample for epigenetic analysis. The baseline characteristics of the caseload and data on AR/ERβ receptors are shown in Table 2 and Table 3, respectively. No significant variations of the investigated serum hormones, with the obvious exception of total testosterone, were detected after GAHT (Figure 1). Appendix A show the variations of the other available clinical parameters during the 12 mo after GAHT. All available clinical and biochemical variables were within normal ranges both at baseline (T0) and during follow-up (T6 and T12). Cessation of menses occurred within the second testosterone vial (around two months). As a measure of successful virilization, we monitored the Ferriman–Gallwey score at baseline and during treatment, detecting a significant increase of the scores at T6 and T12 (*p* < 0.001) (Figure 2).

Post-treatment data from this pilot study showed that the percentage of methylation of *ESR2* increased significantly at T6 (adj. *p* = 0.001) and T12 (adj. *p* = 0.05) (Figure 3), while no significant difference was found in the *H19* promoter methylation (*p* = 0.237).

Linear regressions were performed to evaluate the associations of the *ESR2* methylation variation with available genetic and hormone parameters. Baseline methylation levels were not associated with either AR/ERβ genetic polymorphisms or hormone levels (Table 4). Likewise, no significant associations with *ESR2* methylation were found after six months of testosterone treatment (Table 4). On the other hand, total testosterone and the patient’s age resulted in being significantly associated with *ESR2* methylation after twelve months of GAHT (Table 4). Moreover, although we could not find an association between *ESR2* promoter methylation and the Ferriman–Gallwey score at any time point of the study, the increased virilization of subjects was strongly associated with time from the beginning of treatment (β = 0.918; *p* < 0.001; adj. R^2^ = 0.84) and blood testosterone levels (β = 0.774; *p* < 0.001; adj. R^2^ = 0.55).

Finally, we used linear models to attempt to evaluate predictors of the difference in methylation after starting testosterone treatment. We could detect that the difference in *ESR2* promoter methylation between T6 and baseline was significantly associated with the number of CA repeats of the ERβ receptor, adjusted vs. all considered variables (R^2^ = 0.62, adj. R^2^ = 0.35) (Table 5). On the other hand, only age was significantly associated with the *ESR2* promoter methylation difference between T12 and baseline (R^2^ = 0.73, adj. R^2^ = 0.53) (Table 5).

## 4. Discussion

Gender dysphoria is a relatively rare condition, where the strict link between hormone receptor signaling and exogenous hormone administration is likely to affect clinical outcomes and phenotypical changes. However, the few genetic and epigenetic studies in this setting have focused only on their possible contribution on the development of gender identity/incongruence. A recent metanalysis from D’Andrea et al. (2020) confirmed the presence of a significantly higher number of CAG repeats of *AR* in transgender women compared to cisgender controls [5]. Caution must be taken in interpreting this result as a possible etiological genetic contribution, as it must be remarked that it is still unknown to what extent functional variants of hormone receptor genes might be implicated in gender incongruence. Besides, it is likely that hormonal signaling may be modulated by both congenital and acquired epigenetic factors. Currently, the most characterized epigenetic modification is DNA methylation.

Several articles support the effect of methylation on imprinted genes with growth, growth-related hormone concentrations, adiposity, and birth weight [17,18]. Despite the biological plausibility, we did not observe any evidence supporting an association between H19 promoter methylation and response to GAHT.

Hormone receptor (AR, ERα, and ERβ) promoters’ methylation is involved in the development of many diseases, such as atherosclerosis, endometriosis, and cancer [15,19]. DNA methylation is mainly associated with the function of three enzymes belonging to the DNA methyltransferase family (DNMT1, DNMT3A, and DNMT3B), which maintain or induce de novo methylation of a specific DNA sequence [20]. Fernandez et al. (2020) recently demonstrated different methylation patterns of the *ESR1* gene promoter in cisgender and transgender subjects [7]. Although the authors identified specific methylation patterns in both assigned male at birth (AMAB) and AFAB people before GAHT, the paucity of data does not allow solid conclusions. In fact, it remains to be ascertained whether epigenetic changes in gender incongruence are causative or simply represent an epiphenomenon of other environmental agents.

From a clinical point of view, the main impact from functional variants in hormone receptor signaling is expected to be found both in the timing and amplitude of the phenotypical changes after GAHT. In physiological puberty, hormonal activation is expected to be modulated by the genetic background and the epigenome [21,22]. Although human studies are lacking, environmental influences in the form of the so-called endocrine-disrupting chemicals (EDCs) are also known to impact physiological puberty, possibly affecting the development of sex hormone-dependent organs (external genitalia and other secondary sex characteristics) through interference on both steroidal and non-steroidal hormone receptors either directly or through epigenetic changes [23,24]. Thus, differences in the timing and acquisition of secondary sexual characteristics may depend on differences in epigenomes induced by GAHT and environmental factors. Therefore, epigenetic changes might ultimately affect the response to treatment with sex steroids, even though this hypothesis has not been thoughtfully investigated in the literature. Sader et al. (2005) were among the first to observe a significant downregulation of *AR* mRNA expression in blood after both testosterone and estradiol treatment in transgender subjects (six AFAB and six AMAB, respectively) [25]. More recently, Aranda et al. (2017) reported the effect of exogenous sex steroids on the methylation and expression patterns of hormone receptors [8]. The authors observed a downregulation of *AR* mRNA in the blood of transgender men, as well as an increase in the methylation of *AR* and *ESR1* in transgender women and men, respectively, after 12 mo of treatment. Remarkably, these changes were correlated with several anthropometric (waist circumference), metabolic (HDL, hematocrit), and hormonal (estradiol, testosterone) parameters. This result is particularly important, as it suggests how a fine regulation of the hormone signaling may take place under GAHT. The small sample size of this study does not allow generalizing this observation. Finally, the previously cited study from Fernandez et al. (2020) also analyzed the methylation pattern of *ESR1* after 6 mo of hormonal treatment [7]. The authors found that only transgender men after testosterone treatment had a statistically significant increase of methylation. Similarly, our data showed an increase in methylation of *ESR2* after testosterone treatment of AFAB subjects within a similar time frame (6 mo). Furthermore, these changes appeared to remain constant up to 12 mo. Therefore, it is possible to hypothesize that treatment with exogenous testosterone can modify the pattern of methylation and, possibly, gene expression. However, our data based on a small study population need to be validated on a larger number of subjects in order to establish the role of epigenetic regulation in the modulation of phenotypical changes after GAHT. Nonetheless, our data suggest that epigenetic changes may be strictly regulated by testosterone. In fact, the modulation of estrogen signaling seems relevant within the first six months of treatment, when methylation might work in tandem with CA polymorphisms, as we detected through linear models between baseline and T6, maintaining itself for more prolonged treatments. Age might be involved as it seems to be constantly and positively associated with methylation of the *ESR2* promoter during GATH. In fact, it is known that human ageing is associated with genetic, epigenetic, and environmental factors, which may be, in turn, associated with the development of diseases [26]. Age-dependent modifications in DNA methylations have been already described [27,28], and at least some of these changes might reflect specific environmental exposure of the individual rather than the contribution of genetic factors [29]. Another relevant aspect of this preliminary data stands in the confirmation that estrogen receptor methylation is associated with the administration of exogenous testosterone, while other clinical variables (such as BMI, baseline hormone levels, etc.) seem to play only a marginal role in the induction of epigenetic changes of the *ESR2* promoter. If confirmed on a larger caseload and longer follow-up, we may hypothesize exploiting the modulation of androgenic and estrogenic signaling for GAHT tailoring in transgender men and women. Another potential application of these data could be to use the methylation pattern as a marker of androgenic treatment. Being exogenous androgens are the main determinant of this epigenetic marker at T12, it is possible to speculate that this change may also be present following other forms of exogenous testosterone administration, including doping. Epigenetic modifications have been investigated in the setting of sports medicine, hypothesizing an association between epigenome and physical performance traits [30]. Histone modifications (acetylation, methylation) seem to play a role in exercise adaptation and in particular in the expression of musculoskeletal protein expression [31], although their effects on health and physical performance need to be fully elucidated. As our data indicate an influence of exogenous testosterone administration on the epigenetic markers of *ESR2*, the investigation of these effects in the setting of illegal hormone administration (testosterone, but also other doping agents such as growth hormone, etc.) may be warranted.

In conclusion, data from this pilot study confirm the safety and efficacy of testosterone treatment in AFAB people. *ESR2* promoter methylation is increased after testosterone treatment and remains constant up to one year of treatment. The associations we detected with age and testosterone levels suggest that epigenetic modifications might be modulated from these parameters during treatment. Instead, we could not demonstrate a role for *H19* promoter methylation, a parentally imprinted gene. Future studies are needed in order to explore the sexually differentiated mechanisms behind DNA methylation considering epigenetic regulation in a larger panel of genes that may directly or indirectly influence hormone treatment-related response.

Nonetheless, it is likely that epigenetically modulated hormone signaling may not only affect the timing of phenotype changes, but also may represent a molecular marker of hormone utilization/abuse in both transgender and cisgender people [32]. Current data will assist in the development of studies with larger caseloads to fully elucidate the role of epigenetic modifications in the pharmacological treatment of transgender people.

## Figures and Tables

**Figure 1 biomedicines-10-00459-f001:**
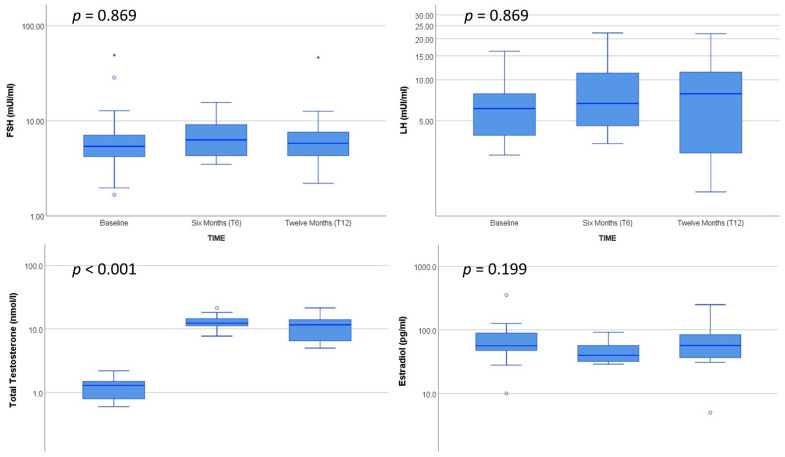
Hormone profile of the recruited subjects at baseline and after 6 mo (T6) and 12 mo (T12) of GAHT. Friedman test. *p*-values are Bonferroni adjusted.

**Figure 2 biomedicines-10-00459-f002:**
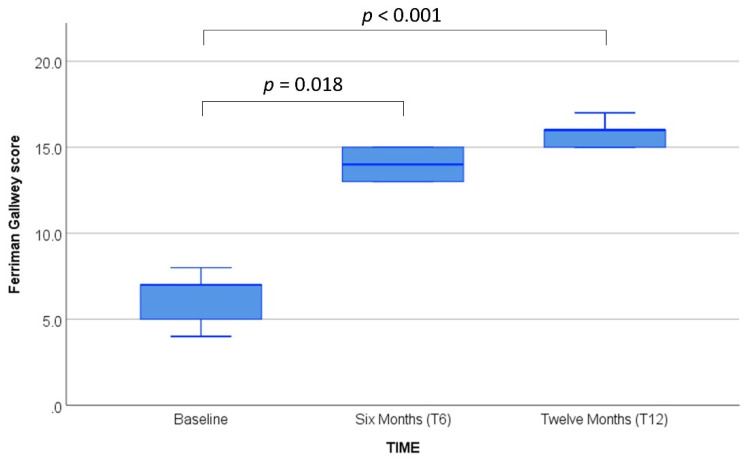
Ferriman–Gallwey score at baseline and after gender assignment hormone therapy (T6 and T12). Friedman’s test. *p*-values are Bonferroni adjusted.

**Figure 3 biomedicines-10-00459-f003:**
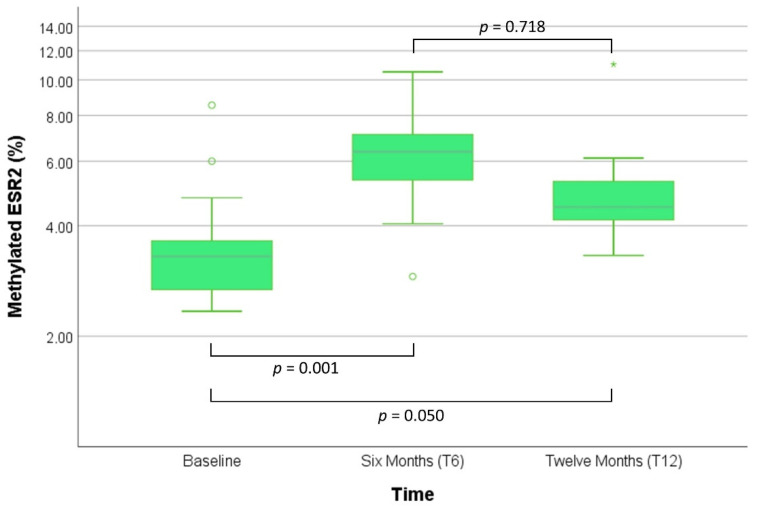
ESR2 methylation percentage at baseline and after gender assignment hormone therapy (T6 and T12). Friedman’s test. *p*-values are Bonferroni adjusted.

**Table 1 biomedicines-10-00459-t001:** DNA methylation primer sequences and PCR cycling conditions.

Primer/Sequence to Analyze	*ESR2*	*H19*
Forward PCR Primer	5′-GGAGGTTGAGAGAAATAATTGTTTTTTGA- 3′	5′-TTTGTTGATTTTATTAAGGGAG-3′
Reverse PCR Primer	5′-[Biotin]-ATAAACACACCCACCTTACCTTCTCTA-3′	5′-[Biotin]-CTATAAATAAACCCCAACCAAAC-3′
Sequencing Primer	5′-GAAATAATTGTTTTTTGAAATTTG-3′	5′-GTGTGGAATTAGAAGT-3′
Sequence to Analyze	TAGGGYGAAGAGTAGGYGGYGAGYGTTGGGTYGGGGAGGGATTATTYGAGTTGYGAYGGGTTTTGGGGTTGYGGGGTA	GGTYGYGYGGYGGTAGTGTAGGTTTATATATTATAGTT
Annealing Temperature (°C)	61 °C for 30 s	51 °C for 30 s

**Table 2 biomedicines-10-00459-t002:** Baseline characteristics of the enrolled AMAB subjects. Continuous data are shown as the mean ± standard deviations and the median (in brackets).

Age (Years)	29.3 ± 12.6 (21)
BMI	26.6 ± 1.4 (27)
CAG repeats expression (AR)	23.1 ± 1.9 (23.5)
CA repeats expression (ERβ)	22.1 ± 1.4 (21.5)
Ferriman–Gallwey score	6.4 ± 1.3 (7.0)
Creatinine (mg/dL)	0.6 ± 0.1 (0.6)
Red blood cells (×10^6^/mL)	4.6 ± 0.5 (4.6)
Hemoglobin (g/dL)	12.7 ± 1.2 (12.6)
Hematocrit (%)	39.0 ± 3.1 (39.9)
White blood cells (×10^3^/mL)	7.1 ± 1.7 (7.1)
Platelets (×10^3^/mL)	243.8 ± 80.3 (252.0)
Glycemia (mg/dL)	85.8 ± 11.8 (85.0)
HbA1c (%)	5.1 ± 0.4 (5.1)
AST (UI/L)	23.5 ± 19.8 (19)
ALT (UI/L)	22.1 ± 10.8 (18.0)
γGT (UI/L)	19.5 ± 12.9 (15.0)
Total cholesterol (mg/dL)	153.8 ± 27.2 (150.0)
HDL (mg/dL)	56.3 ± 11.6 (58.0)
LDL (mg/dL)	80.5 ± 23.1 (85.0)
Triglycerides (mg/dL)	83.8 ± 38.3 (71.0)
FSH (mUI/mL)	10.5 ± 13.5 (5.4)
LH (mUI/mL)	7.8 ± 4.6 (6.1)
Prolactin (ng/dL)	13.2 ± 4.7 (12.2)
17β estradiol (pg/mL)	84.0 ± 86.3 (56.1)
Total testosterone (nmol/L)	1.2 ± 0.5 (1.3)

**Table 3 biomedicines-10-00459-t003:** Polymorphisms of the androgen receptor (CAG repeats) and estrogen receptor beta (CA repeats). For androgen receptor, percentage of X inactivation is provided.

Patient	CAG Repeats (Allele 1)	Percentage of Inactivation (Allele 1)	CAG Repeats (Allele 2)	Percentage of Inactivation (Allele 2)	CA Repeats (Allele 1)	CA Repeats (Allele 2)
#1	24	41.0	25	59.0	24	24
#2	23	62.1	25	37.9	20	24
#3	19	29.6	26	70.4	18	23
#4	17	50.8	24	49.3	20	20
#5	20	35.8	24	64.2	20	23
#6	20	51.3	21	48.7	23	23
#7	24	56.7	26	43.3	18	25
#8	21	56.6	24	43.4	21	24
#9	25	51.4	26	48.6	21	21
#10	22	58.3	25	41.7	17	26
#11	20	/	20	/	24	24
#12	24	45.4	26	54.6	24	24
#13	24	35.8	26	64.2	21	21

**Table 4 biomedicines-10-00459-t004:** Coefficients from regression models predicting baseline ESR2 methylation.

	B	CI 95%	Beta	*p*
Coefficients from regression models predicting baseline ESR2 methylation.				
CA repeats	−0.529	−1.302–0.243	−0.414	0.533
CAG repeats	−0.175	−0.762–0.413	−0.193	0.527
Total testosterone	1.143	−0.852–3.139	0.355	0.233
Estradiol	−0.007	−0.019–0.006	−0.334	0.164
Age	0.026	−0.103–0.155	0.185	0.653
Coefficients from regression models predicting T6 ESR2 methylation.				
CA repeats	0.437	−0.428–1.302	0.318	0.290
CAG repeats	−0.111	−0.751–0.529	−0.114	0.711
Total testosterone	0.030	−0.296–0.357	0.062	0.841
Estradiol	0.009	−0.060–0.077	0.083	0.788
Age	−0.034	−0.130–0.062	−0.230	0.450
Coefficients from regression models predicting T12 ESR2 methylation.				
CA repeats	−0.110	−1.062–0.841	−0.077	0.803
CAG repeats	0.110	−0.560–0.780	0.108	0.725
Total testosterone	0.280	0.086–0.474	0.691	0.009
Estradiol	−0.009	−0.028–0.011	−0.276	0.361
Age	0.106	0.031–0.182	0.682	0.010

**Table 5 biomedicines-10-00459-t005:** Linear regression model coefficients predicting the increase in ESR2 promoter methylation between baseline and T6.

	B	CI 95%	Beta	*p*	Partial Eta Squared
Linear regression model coefficients predicting the increase in ESR2 promoter methylation between baseline and T6.					
CA repeats	1.400	0.357–2.444	0.822	0.016	0.590
CAG repeats	−0.481	−1.385–0.423	−0.400	0.249	0.185
Total testosterone	0.445	−0.061–0.850	0.728	0.076	0.382
Estradiol	0.079	−0.016–0.073	0.609	0.090	0.356
Age	−0.067	−0.183–0.050	−0.361	0.220	0.206
Linear regression model coefficients predicting the increase in ESR2 promoter methylation between baseline and T12.					
	**B**	**CI 95%**	**Beta**	** *p* **	**Partial Eta Squared**
CA repeats	0.498	−0.270–1.266	0.304	0.169	0.251
CAG repeats	0.285	−0.421–0.991	0.246	0.371	0.115
Total testosterone	0.169	−0.179–0.518	0.367	0.289	0.159
Estradiol	0.014	−0.007–0.036	0.407	0.152	0.270
Age	0.109	0.010–0.218	0.615	0.048	0.449

## Data Availability

The datasets used and/or analyzed during the current study are available from the corresponding author upon reasonable request.

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
