# Peer review of "Epigenetic Effects of Gender-Affirming Hormone Treatment: A Pilot Study of the ESR2 Promoter’s Methylation in AFAB People"

_biomedicines, 2022, doi:10.3390/biomedicines10020459_

Round 1
Reviewer 1 Report
Pallotti and collaborators investigated the effects of hormone therapy on ESR2 gene promoter methylation in peripheral blood of assigned female at birth (AFAB) individuals. Authors found that ESR2 methylation increased after 6 and 12 months of treatment, thus suggesting that treatment with exogenous testosterone could modify ESR2 promoter methylation levels. Moreover, correlations between ESR2 methylation and CA repeats of ERβ gene, and age have been detected.
The manuscript is overall well written and results are of interest. I have the following concerns and suggestions.
1) The introduction should be implemented with more information that will allow even less experienced readers to better understand what the purpose of the study is and how the endpoints to investigate were chosen. Maybe some sentences used in the discussion could be moved to the introduction.
2) In section 2.3 authors reported that they investigated H19 promoter methylation, and in table 1 primers and region analysed are also indicated. However, in the results section data on H19 methylation analyses are missing. Authors should clarify this point.
3) In the introduction authors stated that relationship between ESR2 methylation and “acquired body modifications” (line 73) will be investigated. However, in the results section no data on this relationship has been reported. Is it possible to obtain information on the success of the testosterone treatment in the 13 AFAB individuals enrolled, and if yes, they could be quantified? Correlations of these markers with ESR2 methylation will greatly improve the manuscript.
4) In table 2, the mean baseline Ferriman–Gallwey score is reported. Is this data also available for T6 and T12? It should be of interest to correlate such score with ESR2 methylation.
Minor comments.
Lines 39-40: What authors mean with “…containing itself for more prolonged treatments”?
Line 108: in the sentence “… a volume of di 25….”, “di” should be deleted.
In table 5.a and 5.b “ER repeats” should be replaced with “CA repeats”.
Author Response
Reviewer 1
Pallotti and collaborators investigated the effects of hormone therapy on ESR2 gene promoter methylation in peripheral blood of assigned female at birth (AFAB) individuals. Authors found that ESR2 methylation increased after 6 and 12 months of treatment, thus suggesting that treatment with exogenous testosterone could modify ESR2 promoter methylation levels. Moreover, correlations between ESR2 methylation and CA repeats of ERβ gene, and age have been detected.
The manuscript is overall well written, and results are of interest. I have the following concerns and suggestions.
Q1: The introduction should be implemented with more information that will allow even less experienced readers to better understand what the purpose of the study is and how the endpoints to investigate were chosen. Maybe some sentences used in the discussion could be moved to the introduction.
A1: we thank the reviewer for his suggestion. We have improved the introduction section in light of both reviewers’ insightful comments.
Q2: In section 2.3 authors reported that they investigated H19 promoter methylation, and in table 1 primers and region analysed are also indicated. However, in the results section data on H19 methylation analyses are missing. Authors should clarify this point.
A2: Data on H19 was not presented due non-significant associations with treatment. We agree with the reviewer that this aspect should not have been avoided. Thus we revised the manuscript adding the requested information.
Q3: In the introduction authors stated that relationship between ESR2 methylation and “acquired body modifications” (line 73) will be investigated. However, in the results section no data on this relationship has been reported. Is it possible to obtain information on the success of the testosterone treatment in the 13 AFAB individuals enrolled, and if yes, they could be quantified? Correlations of these markers with ESR2 methylation will greatly improve the manuscript.
Q4: In table 2, the mean baseline Ferriman–Gallwey score is reported. Is this data also available for T6 and T12? It should be of interest to correlate such score with ESR2 methylation.
A3 and A4: We thank the reviewer for these comments. We do have some data available for the virilization status of the subjects, which can be objectively reported using the FG score which have been added into the manuscript. In general, in this small cohort, cessation of menses in all subjects occurred within the second testosterone vial (2 months of treatment, median 1 month) and FG scores significantly improved after 6 months of treatment. This was significantly correlated with testosterone treatment, but no specific correlation was detected with genetic polymorphisms or ESR2 promoter methylation. The result section has been amended in light of your suggestions.
Minor comments.
Q5: Lines 39-40: What authors mean with “…containing itself for more prolonged treatments”?
A5: We meant that ESR2 methylation did not modify significantly between T6 and T12. The text has been clarified.
Q6: Line 108: in the sentence “… a volume of di 25….”, “di” should be deleted.
Q7: In table 5.a and 5.b “ER repeats” should be replaced with “CA repeats”.
A6-7: The highlighted inaccuracies have been corrected. Thanks.
We finally pose to the reviewer’s attention that the Material and methods section was slightly modified based on editor’s suggestion.
Reviewer 2 Report
Polymorphisms of sex hormones receptors and methylation of
their gene promoters were studied in adults with gender incongruence. Resuts may be usful for extension of results in further research
1. In Abstract please use full names of used abbreviations and check it throughout ms
2. Introduction is too short -not much information is provided on adults with gender incongruence
3. “body modification” -I am not sure if this description is correct in terms of manuscript content
4. In Results there is information that 13 AFAB patients were examined. I think that there should be reference to healthy patients (in term of studied parameters)
5. Conclusion is poor it should be improved
Author Response
Reviewer 2
Polymorphisms of sex hormones receptors and methylation of their gene promoters were studied in adults with gender incongruence. Results may be useful for extension of results in further research
Q1: In Abstract please use full names of used abbreviations and check it throughout ms
A1: We are sorry for incomplete specifications of abbreviations used. This has been checked and corrected through the manuscript.
Q2: Introduction is too short -not much information is provided on adults with gender incongruence
A2: we thank the reviewer for his comment. We have improved the introduction section in light of both reviewers’ suggestions.
Q3: “body modification” -I am not sure if this description is correct in terms of manuscript content
A3: The manuscript has been amended, as requested.
Q4: In Results there is information that 13 AFAB patients were examined. I think that there should be reference to healthy patients (in term of studied parameters)
A4: All available clinical and biochemical variables were within normal ranges both at baseline (T0) and during follow up (T6-T12). This has been added to the manuscript.
Q5: Conclusion is poor it should be improved
A5: The conclusive remarks have been improved as suggested.
We finally pose to the reviewer’s attention that the Material and methods section was slightly modified based on editor’s suggestion.
Round 2
Reviewer 1 Report
The manuscript has been improved.
Reviewer 2 Report
Author implicated all suggestions into the ms corrections